# Associations between Pregnane X Receptor and Breast Cancer Growth and Progression

**DOI:** 10.3390/cells9102295

**Published:** 2020-10-15

**Authors:** Bradley A. Creamer, Shelly N. B. Sloan, Jennifer F. Dennis, Robert Rogers, Sidney Spencer, Andrew McCuen, Purnadeo Persaud, Jeff L. Staudinger

**Affiliations:** Division of Basic Sciences, College of Osteopathic Medicine, Farber-McIntire Campus, Kansas City University, Joplin, MO 64804, USA; bcreamer@kansascity.edu (B.A.C.); ssloan@kansascity.edu (S.N.B.S.); jdennis@kansascity.edu (J.F.D.); rrogers@kansascity.edu (R.R.); sspencer@kansascity.edu (S.S.); andrew.mccuen@kansascity.edu (A.M.); narpaul@kansascity.edu (P.P.)

**Keywords:** nuclear receptor, pregnane X receptor, xenobiotics, tumorigenesis, breast cancer

## Abstract

Pregnane X receptor (PXR, NR1I2) is a member of the ligand-activated nuclear receptor superfamily. This receptor is promiscuous in its activation profile and is responsive to a broad array of both endobiotic and xenobiotic ligands. PXR is involved in pivotal cellular detoxification processes to include the regulation of genes that encode key drug-metabolizing cytochrome-P450 enzymes, oxidative stress response, as well as enzymes that drive steroid and bile acid metabolism. While PXR clearly has important regulatory roles in the liver and gastrointestinal tract, this nuclear receptor also has biological functions in breast tissue. In this review, we highlight current knowledge of PXR’s role in mammary tumor carcinogenesis. The elevated level of PXR expression in cancerous breast tissue suggests a likely interface between aberrant cell division and xeno-protection in cancer cells. Moreover, PXR itself exerts positive effect on the cell cycle, thereby predisposing tumor cells to unchecked proliferation. Activation of PXR also plays a key role in regulating apoptosis, as well as in acquired resistance to chemotherapeutic agents. The repressive role of PXR in regulating inflammatory mediators along with the existence of genetic polymorphisms within the sequence of the PXR gene may predispose individuals to developing breast cancer. Further investigations into the role that PXR plays in driving tumorigenesis are needed.

## 1. Introduction

Pregnane receptor (PXR) (NR1I2), termed steroid and xenobiotic receptor (SXR) in humans, is a nuclear receptor that acts as a xenobiotic and endobiotic sensor that recognizes a large variety of hydrophobic ligands [1]. PXR is highly expressed in small intestine, liver, rectum, colon, and gallbladder, but is also expressed in comparatively lower levels in reproductive tissues including breast tissue [2,3]. First discovered in 1998, PXR is a key regulator of the expression of cytochrome-P 450 (CYP) enzymes, including CYP3A4 and CYP2B6 [4,5,6,7,8], which metabolize a variety of toxins, carcinogens, and clinically relevant chemotherapeutic agents (i.e., tamoxifen and doxorubicin) [1,9,10,11,12,13]. The role of PXR as a nuclear receptor has evolved to also include regulation of families of genes involved in oxidative stress, steroid and bile acid metabolism, inflammation, apoptosis, cell proliferation, and cell cycle maintenance [14,15,16,17,18]. Activation of PXR regulates key aspects of cancer cell biology to include tumor metastasis, angiogenesis, acquisition of chemotherapeutic resistance, and the inflammatory immune response [1,2,19].

PXR activity is governed by the coordinated efforts of ligands, other nuclear receptors and cofactors, as well as epigenetic and post-translational modifications such as phosphorylation, acetylation, ubiquitination, and SUMOlytion. The DNA-binding activity of PXR is activated by a wide variety of endobiotic compounds including steroid hormones and bile acids such as estrogen or lithocholic acid, respectively. PXR is also activated by a plethora of exogenous compounds including xenobiotic chemicals, food ingredients, and a wide variety of herbal remedies and pharmaceutical agents. The net effect of canonical ligand-mediated PXR activation in mammals is the coordinate regulation of a pathway of genes that encode pivotal hepatic and intestinal proteins which are involved in the coordinate uptake, biotransformation, and eventual excretion of a myriad of specific—and potentially toxic—metabolites in feces or urine [1].

The activation of PXR regulates the expression of hepatic uptake drug and bile acid transporter proteins including organic anion transporting polypeptide protein 2 (Oatp2) and organic cation transporter 1 (Oct1) [20,21,22]. Multi-drug resistance-associated protein 2 (MRP2) and MRP3, both of which are responsible for efflux on conjugated substrates into bile or blood, respectively, are proteins encoded by PXR-target genes [22,23,24]. In the intestine, another PXR-target gene is multi-drug resistance 1 (MDR1) that encodes P-glycoprotein 1 (P-gp) [25,26], another broad selective drug efflux membrane transporter. Because PXR is important for the uptake and efflux of steroid hormones and metabolites due to the variety of transport proteins that are encoded by PXR-target genes, its ability to link endocrinology, pharmacology, and cancer biology is essential to maintaining cellular homeostasis. 

Given the breadth of compounds that activate PXR and its role in the coordination between multiple biological processes, it makes sense that there would be crosstalk between PXR and other nuclear receptors including farnesoid X receptor (FXR), constitutive androstane receptor (CAR), peroxisome proliferator-activated receptor alpha (PPARα), liver X receptor (LXR) and androgen receptor [2]. In addition, like other nuclear receptors, PXR activity is modulated by a number of cofactors further placing PXR as a nexus of molecular signaling. The PXR corepressors, nuclear receptor corepressor 1 (NCoR1) and NCoR2 (also known as silencing mediator of retinoid and thyroid hormone receptors, SMRT), are bound to inactive PXR until an activator is present. Additional PXR corepressors include sterol regulatory element binding protein 1 (SREBP-1) and forkhead box A 2 transcription factor (FOXA2) [2,27,28]. Activators of PXR, which include some anticancer drugs, induce a conformational change resulting in dissociation of corepressors and recruitment of coactivators [19]. Coactivators that have been identified include steroid receptor coactivators (SRCs) 1, 2 and 3, peroxisome proliferator activated receptor gamma coactivator 1-alpha (PGC-1α), forkhead box O 1 transcription factor (FOXO1), protein arginine methyltransferase (PRMT) and p300 [2,29,30]. Thus, activity of PXR can be influenced by a number of factors including and excluding direct activators or inhibitors. In this review, we will summarize the current knowledge of the mechanisms of PXR’s role in breast cancer. PXR is an intriguing molecule in cancer because it has been observed to play a role in apoptosis, oxidative stress, cell cycle arrest, tissue growth, proliferation, and tumor aggressiveness, as well as serving as a marker of poor prognosis in cancer patients [14,15,31,32,33,34]. PXR also is involved in metabolism of estrogen and estrogen metabolites, in addition to chemotherapeutic drugs making it a potential therapeutic target.

## 2. PXR and Breast Cancer

### 2.1. PXR and the Cell Cycle

The primarily anti-apoptotic role of PXR in the cell cycle and cell proliferation has been evidenced through research on hepatocytes. It has long been established that PXR activation in hepatocytes can induce liver regeneration and even lead to hepatomegaly. PXR was specifically found to do this through an interaction with YAP [35]. Shizu et al. (2013) also demonstrated this in a study which showed that PXR stimulation with PCN induced proliferation in liver cells. PXR has also been found to induce proliferation of human breast cancer cells [31,36,37,38,39]. It was further shown that the treated breast cancer cells had decreased levels of p27 and p130 mRNA, both of which inhibit entry of quiescent cells into the cell cycle [40]. Overall, these studies point toward PXR causing increased cell cycle progression in hepatocytes and potentially human breast cancer.

The relationship of p53 and PXR is likely also important in PXRs role in cancer. The tumor suppressor p53 is essential for inducing cell cycle arrest in DNA damaged cells and is often referred to as the ‘guardian of the genome’. Without activated p53, the cell cycle goes unchecked predisposing it to malignancy. Recent studies have demonstrated that PXR exerts an inhibitory effect on p53 through reducing p53 transactivation along with the expression of target genes involved in apoptosis and cell cycle arrest [41]. Alternately, activated p53 suppresses PXR and therefore drug metabolism, which can enhance chemotherapeutic effects [42]. 

Another study provided further evidence for the antiapoptotic function of PXR independent of its drug detoxifying effects through the upregulation of antiapoptotic proteins and downregulation of proapoptotic proteins. In the study, PXR-producing colon cancer cells were exposed to deoxycholic acid, a component of bile acid capable of inducing apoptosis in colon epithelium, both in the presence and absence of a PXR agonist. Expression of antiapoptotic genes—*BAG3*, *BIRC2*, and *MCL-1*—were subsequently found to be elevated while proapoptotic genes BAK1 and p53 were decreased. Similar results were observed in normal mice colon epithelium with activation of PXR leading to decreased deoxycholic acid-induced apoptosis and increased sensitization to colon cancer [15]. While PXR has been linked to some anti-apoptotic mechanisms in breast cancer, studies have indicated that PXR induces apoptosis in the p53 wild-type breast cancer MCF-7 and ZR-75-1 cell lines following upregulated expression of p21, PUMA, and BAX [43]. PXR knockdown using targeted siRNA was able to block the PXR-induced apoptosis in MCF-7 cells, indicating a regulatory role of apoptosis induction in breast cancer cells [43].

### 2.2. PXR and Chemotherapeutic Resistance

Acquired resistance to chemotherapies in tumor cells is a complex process, with multiple genes likely involved. Changes in gene expression of multidrug resistant-associated proteins is thought to play a major role in the transition from responsive to resistant cancers. Several downstream target genes of PXR may be involved in this transition, including phase I metabolic enzymes, CYP enzymes, phase II metabolic enzymes, glucuronyltransferase (UGT) and phase III drug transporters, multidrug resistance protein 1 (MRP1), and breast cancer resistance protein (BCRP) [44,45].

Although PXR is expressed mainly in the liver, intestine, and colon tissues, it has been found to be expressed in normal breast tissue, and at even higher levels in breast cancer tissue [31,36,37,38,39,43]. In addition, PXR expression has been shown to be induced in breast cancer cell lines MDA-MB-231 and MCF-7 in response to PXR ligand administration [46,47]. Forced overexpression of PXR in these cells resulted in an increased promoter activity and cellular level of drug resistance proteins such as MRP1 and BCRP [48,49]. MDR1 and BCRP are members of the ATP binding cassette (ABC) superfamily of transporters, and function to increase the cellular outflow of many types of chemotherapeutic drugs [50]. When overexpressed, or treated with the PXR agonist, SR12813, PXR reduced the response in breast cancer cells to tamoxifen, cisplatin, and paclitaxel treatment, whereas its downregulation restored cell cycle regulation and apoptosis [47,48]. Li et al., in 2016, provided evidence that the protein arginine methyltransferase 1 (PMRT1), a known epigenetic modifier of histone H4, is an important co-activator of PXR and aids in driving the expression of the MDR1 gene during acquired chemoresistance [46]. In addition, significant correlations between BCRP expression and resistance to 5-FU treatment has been noted [51], as well as indications that PXR may modulate a TGF-β induced resistance to chemotherapy in liver cells, via alterations to procaspase-3 and Mcl-1 levels [52].

The drug uptake transporter organic anion transporter polypeptide 1A2 (OATP1A2), which mediates cellular uptake of estrogen metabolites, has also been documented as a downstream target of PXR. OATP1A2 was first identified through its characterization of enhanced expression in breast cancer as compared to healthy breast tissues [39]. Meyer zu Schwabedissen and colleagues (2008) demonstrated, in T47-D cells, the activation of PXR through agonist treatment induces OATP1A2 expression in a time- and dose-dependent manner, resulting in enhanced estrogen receptor activation [38]. The authors identified and confirmed a PXR response element in the human OATP1A2 promoter 5.8 kilo bases upstream of the transcription initiation site. This provided insight into the interactions of PXR and its downstream target, which may contribute to breast cancer pathogenesis and development of resistance towards chemotherapy in breast cancers expressing PXR.

### 2.3. PXR and Metastasis and Angiogenesis

Cancer cell motility leading to metastasis, along with angiogenesis, are major components of cancer progression and areas in which PXR may be involved. PXR, when activated, has been shown to be sufficient at enhancing neoplastic characteristics including: cell growth, invasion, and metastasis through activation of Fibroblast Growth Factor 19 (FGF19) gene expression in colon cancer. Shown in both human colon tumor cell lines and in primary human colon cancer tissue xenografted into immunodeficient mice, the role of PXR activation through FGF19 induction mediates colon cancer cell proliferation and migration [14]. PXR’s implication for playing a role in human cancer metastasis may also stem from it being shown to regulate morphology and cell motility via the signaling axes of PXR GADD45β-p38MAPK and PXR-HNF4ɑ-insulin-like growth factor-binding protein 1 (IGFBP1) [53,54]. Using hepatocellular carcinoma cells (HepG2) to investigate cell motility, it has been shown that PXR activation leads to increased phosphorylation of p38 MAPK through GADD45β gene induction. It was then observed that the HepG2 cells underwent epithelial-to-mesenchymal morphologic changes leading to reorganization of actin filaments and increased cell migration [53]. This PXR-dependent regulation of GADD45β and IGFBP1 helps support its diverse role in cellular regulation since both genes are found in cellular processes including: apoptosis, the cell cycle, DNA repair, cell proliferation, and cell migration [55]. These results can be correlated to a similar study on breast tissue done in 2002 by Bakin et al., who demonstrated that the p38 MAPK phosphorylation pathway is necessary for TGFβ to induce cell migration in both tumor and non-tumor mammary epithelial cells [56]. This points to a possibility that PXR can have similar cell migration-inducing effects in breast tissue as those observed in HepG2 cells, although further research is necessary. Conversely, in a recent study on the effects of a microbiome metabolite, indolepropionic acid (IPA), on PXR activation, Sari et al. (2020) showed that patients with higher expression of PXR have increased survival rates in estrogen receptor-positive cancers [57]. Furthermore, IPA supplementation in mice challenged with 4T1 breast cancer cells showed reduced localized infiltration and metastases [57]. This suggests that PXR may target genes and pathways that reduce breast cancer progression.

Another important factor that contributes to cancer growth and spread is angiogenesis. This process is primarily mediated by vascular endothelial growth factor (VEGF) and nitric oxide (NO). This process works first by the release of VEGF, which then activates the Nuclear Factor Kappa B (NF-κB) pathway leading to an increase in nitric oxide synthase and thus nitric oxide. Nitric oxide then stimulates angiogenesis of new blood vessels needed by cancer cells to thrive [58]. Esposito et al. (2016) found that PXR activation could inhibit this process via its inhibition of the NF-κB pathway leading to decreased levels of NO. Specifically, they found that when Caco2-human colon cancer cells were treated with a non-absorbable antibiotic and PXR agonist rifaximin, there was a decrease in VEGF, NO, VEGFR-2, MMP-2, and MMP-9 along with inhibition of HIF-1ɑ, P-70S6K, and NF-κB. These effects were not observed in cells co-treated with the PXR antagonist ketoconazole [58]. These results indicate a possible inverse relationship between PXR activation and angiogenesis.

### 2.4. PXR and the Inflammatory Immune Response

Historically immune responses, such as inflammation, were thought to help eradicate tumors but evidence is growing that antitumoral responses put pressure on the tumor to evade immune destruction. The intersections between inflammation and cancer pathogenesis has shown the functionally important tumor-promoting effects that immune cells in the innate system have on cancer progression [59]. It has long been established that chronic inflammation can be a predisposition to cancer in a variety of settings. One example being the increased risk of colon cancer in patients with chronic inflammatory bowel disease (IBD). This is thought to be mediated by proinflammatory factors which can induce neoplastic mutations, resistance to apoptosis, and even stimulate angiogenesis [60]. Other studies discuss an inflammatory component present in the microenvironment of tumors that contributes to proliferation and survival of malignant cells, angiogenesis, metastasis, subversion of adaptive immunity, and reduced response to hormones and chemotherapeutic agents known as cancer-related inflammation (CRI) [61]. Transcription factors such as NF-κB and signal transducer activator of transcription-3 (Stat3) and primary inflammatory cytokines, IL-1B, IL-6, and TNF-α are the primary players involved in CRI. Inflammatory cytokines are known to affect PRX regulation either directly or indirectly by modifying transcription factor expression and function such as in STAT3 and NF-κB in the liver, both transcription factors of PXR [27,62,63]. There is substantial evidence that inflammation, with or without infection, decreases the expression of enzymes involved in xenobiotic metabolism such as cytochrome P450 enzymes, however, the underlying mechanism behind this is not clear. For this reason, PXR is thought to likely play a role in the progression of chronic inflammatory pathways. [64]. Lipopolysaccharide (LPS) treatment, a bacterial endotoxin administered to cells or rodents to induce inflammatory changes, has shown to downregulate the mRNA or protein levels, of both PXR and CYP [27]. Inflammatory cytokines: IL-1B, IL-6, and TNFα, when upregulated following LPS treatment, lead to repression of PXR and PXR-mediated CYP induction during inflammation [27]. Elevated levels of cytokine expression following PXR deletion in mice suggests PXR plays a repressive role in regulating key inflammatory mediators [65]. This data suggests there is negative regulation by LPS or LPS-induced cytokines providing a molecular mechanism for inflammation impaired drug metabolism. Protein kinases such as PKA and PKC, who become upregulated during inflammation, also affect PXR and ultimately CYP regulation through transcriptional suppression.

Cancer-induced inflammation is present in at least 60% of patients with advanced cancer and should be considered as a source of variability in the clearance of anticancer drugs [62]. Specifically the role of CYP3A4, which is responsible for metabolizing many anticancer drugs, has been shown to have significantly reduced activity when increased levels of inflammatory mediators are present such as IL-6 and C-reactive protein (CRP) [63,66]. The association of reduced CYP3A4 expression in the presence of cancer is associated with the inflammatory response and the release of cytokines such as IL-6 from the cancer into the blood. The increased circulating CRP and IL-6 along with reduced CYP3A4 activity observed in a subset of cancer patients indicates the role of IL-6 as being a potential mediator in the process [63]. Chronic inflammation has also been implicated in anorexia-cachexia, a condition about half of all cancer patients suffer from [67]. It is characterized by lack of response to treatment, substantial weight loss, and poor prognosis. Though not as common in hematological malignancies and breast cancer, patients with anorexia-cachexia have been shown to have raised serum concentrations of IL-6 and CRP, again associating these inflammatory markers with survival in malignant disorders including myeloma, melanoma, ovarian cancer, renal-cell carcinoma, and gastrointestinal cancers [66,67].

Recent studies have found a link between PXR and the proinflammatory mediator NF-κB. These studies actually demonstrated an inverse relationship with activated PXR inhibiting NF-κB signaling. It was also found that NF-κB target gene expression was upregulated in PXR knockout mice leading to a proinflammatory state [64]. NF-κB was shown to downregulate both PXR and PXR-mediated CYP expressions through protein–protein interactions of the PXR pathway [27,68]. Gu et al. (2006) reported NF-κB activation by LPS or TNF-α suppressed PXR by interacting with the PXR-RXR heterodimer and the suppressive effects could be reversed by the NF-κB specific suppressor SRIκBa [68]. Additionally, inhibition of NF-κB potentiating PXRs activity was also shown by Zhou et al. (2006) [64]. Both are examples of the PXR-NF-κb axis providing a molecular explanation for CYP expression to be suppressed by inflammatory stimuli in hepatic and interstitial tissue. Less is known about STAT3 and PXR activity during inflammation though it has been shown IL-6 activates STAT3 which inhibits the transcriptional activity of other nuclear receptors (NR) such as NNF4a and HNF4a-mediated CYP induction [27,62,63]. The anti-inflammatory role of PXR is further supported by evidence that PXR is downregulated in patients with IBD and several polymorphisms that lead to decreased PXR activity have been found; both have been linked to the presence of IBD in patients [69,70].

### 2.5. Role of PXR Polymorphisims

Genetic polymorphisims within the sequence of PXR, including several SNPs within the 3′UTR, have been suggested to correlate with an increased predisposition to breast cancer by influencing its expression and/or function [48,71]. Additionally, alterations in expression of target genes including ABCB1 and CYP3A4 in breast cancer patients, which subsequently results in altered clearance of xenobiotics and drug substrates, have been linked with PXR polymorphisms [72,73].

A number of SNPs within genes involved in the endobiotic metabolism of estrogen have been suggested to alter breast cancer susceptibility, through modification of circulating estrogen levels. A recent meta-analysis was performed on 10 studied SNPs in PXR and the susceptibility to overall cancer [71]. Two of the 10 (rs3814058 C/T and rs3814057 A/C) were demonstrated to be associated with an elevated risk of cancer. For the SNP rs3814058 C/T, heterozygote and homozygous variants (dominant, recessive, and allelic models) were all correlated with an elevated risk of breast, lung, and colorectal cancer in Asian populations. Additionally, for the SNP, rs3814057 A/C, the heterozygote genotype and all dominant models were found to be correlated with an increased breast and colorectal cancer risk in the entire patient population reviewed as part of the meta-analysis (patient home countries of Germany, China, Malaysia, India, and Mexico). However, meta-analysis data specific to rs3814057 revealed three case-control studies, but none reported an associated cancer risk with rs3814057 [74]. Indeed, the MARIE-GENICA Consortium on Genetic Susceptibility for Menopausal Hormone Therapy Related Breast Cancer Risk reported a decreased risk of breast cancer associated with SNPs rs6785049 and rs1054191, in carriers who used combined estrogen–progesterone, post-menopausal hormone therapy [75]. Seven specific PXR SNPs were analyzed in this patient population.

Revathidevi et al. (2016) screened the 3′ UTR of PXR in 96 breast cancer patients to identify polymorphisms that impact PXR expression levels [48]. They found three SNPs (rs3732360, rs1054190, and rs1054190) which were shown to alter miRNA mediated regulation of PXR. In silico analysis of these three SNPs revealed dramatic alterations to miRNA regulation; 9 new binding sites were identified and 11 sites were eliminated in the 3′ UTR of PXR. The efficiency of selected binding sites was impacted as well: three miRNA sites resulted in increased binding efficiency while two sites were diminished. Furthermore, altered miRNA-binding sites were evaluated for their involvement in tumor progression and toxicity, and were associated with breast cancer pathogenesis, cardiotoxicity, and treatment response [48].

## 3. Conclusions

PXR is expressed and is biologically functional in breast tissue, and a preponderance of evidence suggests that it likely plays a key role in acquired chemoresistance. PXR effects on tissue growth, metastasis, and apoptosis are very likely a tissue-specific phenomenon, whereby in one tissue it may act as a growth proliferator (e.g., colon) but in others (e.g., breast) it could potentially have an opposite or variable overall effect. There is good precedence for this notion as in enterocytes, PXR activation regulates the drug-inducible expression of the *MDR1* gene, whereas in liver it does not [25,26,76]. This phenomenon is almost certainly due to tissue-specific and differential repertoire of nuclear receptor coregulatory proteins available in the given cell type. Similar paradigms exist for several other nuclear receptors. For example, the androgen receptor in breast cancer behaves differently than in the prostate [77,78].

PXR has been shown to be moderately expressed in both ER+ and ER- breast tumors [79], while others report PXR expression is inversely correlated with the expression of ER, suggesting that PXR-mediated pathways might be more active in breast tumors which are less likely to respond to endocrine therapies [37]. A more recent study suggests there is concomitant expression of PXR and ER, with higher PXR expression leading to increased survival rate in ER+ tumors [57]. The range of effects of PXR based solely on ER status strongly suggest that further exploration into PXR and its role within the defined molecular subtypes of breast cancers is warranted. The role of PXR and other nuclear receptors in breast cancer provide an extraordinary amount of complexity within transcriptional regulation and extensive crosstalk, which is likely to be confounded within the multiple molecular subtypes of breast cancer. It is clear that nuclear receptors, although highly dependent upon subtype, have a number of functional roles in breast cancer. In particular, investigating gene regulatory networks under the control of PXR and other nuclear receptors will be crucial in elucidating the role of these transcriptional regulators in breast cancer therapy.

Furthermore, the idea of PXR functionality in breast cancer initiation and breast cancer development is an area that needs further investigation. Differences in the role of PXR based on tumor maturity would most likely be due to the presence or absence of co-factors and target pathways. Additionally, it is likely that modifications of PXR through post-transcriptional and -translational modifications—as well as splice variants and polymorphisms—are associated with breast cancer risk, development, and outcome. Further investigations into the role that PXR plays in context-specific physiological processes via target gene expression, the relevance of its interactions with additional pathways to alter cellular responses, as well as various regulatory mechanisms that may be altered due to structural modifications to PXR are needed.

It is highly possible that PXR has dual roles in both growth and the inhibition of growth. For example, PXR in early cancer development could function in an opposite manner when compared in late cancer activation. This kind of behavior is seen with inflammasomes (e.g., NLRP3) [80,81,82,83]. Differences in outcome-specific PXR activation is also possible, for example, PXR inhibits inflammation and inflammation-induced colitis [84], but promotes late sporadic colon cancer [15]. Could this be true in other tissues like the breast (e.g., inflammatory breast cancer vs. other types, or hormone negative vs. hormone positive)? The future thrust of research efforts should examine PXR involvement in breast cancer to decipher the temporal role of PXR through tumor initiation, progression, and eventual metastases. We believe that drugs that target PXR to activate or inhibit its transactivation capacity could be used in diverse experimental model systems of breast cancer to further unravel any therapeutic opportunities targeting this important nuclear receptor superfamily member. (Figure 1).

## Figures and Tables

**Figure 1 cells-09-02295-f001:**
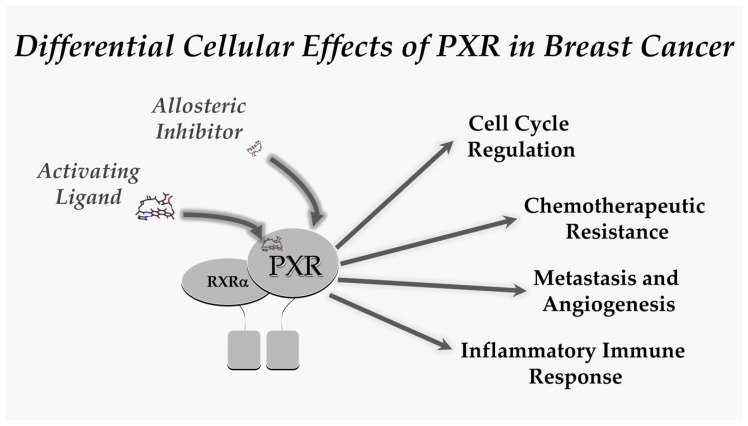
The scheme depicts the pharmacological tools available to examine the therapeutic potential that exists in the field of PXR-mediated modification of breast cancer initiation, progression and metastases. We believe that this represents an important and exciting opportunity to move this field forward.

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
