# Peer review of "Associations between Pregnane X Receptor and Breast Cancer Growth and Progression"

_cells, 2020, doi:10.3390/cells9102295_

Round 1

Reviewer 1 Report

The overall review is appropriate for the receptor PXR in breast cancer - there are very few out there in the literature. The conclusions paragraph could potentially highlight the summary a bit better:

  1. PXR is expressed and functional in breast tissue and the preponderance of the evidence suggests that it has a role in chemoresistance.
  2. PXR effects on tissue growth, metastasis, and apoptosis are likely a tissue-specific phenomenon, whereby in one tissue it may act as a growth proliferator (e.g., colon) but in others (e.g., breast) it could have an opposite effect. This is not unprecedented as several other nuclear receptors behave in a similar manner - androgen receptor (AR) in breast cancer behaves differently than AR in the prostate.
  3. It is highly possible that PXR has dual riles in both growth and inhibition of growth - for example, PXR is early cancer development could function in an opposite manner when compared in late cancer activation. This kind of behavior is seen with inflammasomes (e.g., NLRP3). 
  4. Differences in concept specific PXR activation is also possible- for example, PXR inhibits inflammation and inflammation-induced colitis but promotes late sporadic colon cancer. Could this be true in other tissues like the breast (e.g., inflammatory breast cancer vs other types, or hormone negative vs hormone positive)? 

I think spelling these points out clearly with references in the statement of the conclusion gives the reader a direction as to where the field is headed and what needs to be deciphered.

Reviewer 2 Report

This review deals with the PXR and its suggested significance in breast cancer. The review is concise but still manages to review several aspects of breast cancer biology and their relation with PXR action.

The review is timely and is a good fit for the readership of this journal. The review is mostly well-written. 

My main point is that there is no mention of different histological and molecular subtypes of breast cancer in relation to PXR expression and action. As ductal and lobular breast cancers and ER, PR and HER2 expressors/non-expressors behave and are treated differently, it warrants the authors to include some information on what is known about the differences of PXR expression and action in these subtypes. This topic is alluded to in the Conclusions but not really discussed in the review.

In addition, perhaps a table or figure would be a good addition to the review? 

Minor points include:

Line 9, x should be capitalized

Line 14: word "in" is missing

Lin 116: should be "CYP" not "CYP450"

Line 120: reference missing for the statement that PXR is highly expressed in breast cancer tissue

LIne 216: incomplete sentence

Line 233: typo PRX

Line 237: period missing in the end of sentence, and the sentence itself is unclear.

Line 240 "that" missing
